# How long do new medicines take to reach Canadian patients after companies file a submission: A cohort study

Joel Lexchin[1,2,3]*

1 School of Health Policy and Management, Faculty of Health, York University, Toronto, Ontario, Canada,
2 Emergency Department, University Health Network, Toronto, Ontario, Canada, 3 Department of Family and Community Medicine, Faculty of Medicine, University of Toronto, Toronto, Ontario, Canada

* jlexchin@yorku.ca

**Data Availability Statement:** All relevant data are within the paper and its Supporting Information files.

**Funding:** The author received no specific funding for this work.

## Abstract

### Introduction

Studies of the delay between when companies file a New Drug Submission (NDS) and when drugs reach Canadian patients typically focus on the time in the regulatory review process and do not analyze the time between when approval is granted and the drug is available for purchase (company decision time). This study looks at the length of the two different time periods. Secondarily, it examines whether there is a difference in these time periods for drugs that received a standard review and those that received an expedited review.

### Methods

A list of all New Active Substances approved in Canada between January 1, 2014 and December 31, 2018 was compiled and the dates when the companies applied for a NDS, the dates when the drugs received a market authorization (Notice of Compliance, NOC) and whether the drugs received a standard review or an expedited review were recorded. The date of original marketing comes from Health Canada's Drug Product Database. Times in days were calculated between NDS and NOC (review time), between NOC and the marketing date (company decision time) and between NDS and the marketing date (total time). The company decision time as a percent of the total time was calculated for all drugs. Times were compared between standard and expedited review drugs using a two-tailed t-test.

### Results

One hundred and fifty-seven drugs were analyzed, 98 had a standard review and 59 had a priority review. Over 18% of the total time was due to company decisions. All three times were significantly lower for expedited review drugs versus standard review drugs as was the percent of total time due to company decision– 14.4% (95% CI 11.0, 17.8) versus 21.2% (95% CI 17.6, 24.8), p = 0.0102 (t-test).

**Competing interests:** In 2016-2019, Joel Lexchin was a paid consultant on two projects: one looking at developing principles for conservative diagnosis (Gordon and Betty Moore Foundation) and a second deciding what drugs should be provided free of charge by general practitioners (Government of Canada, Ontario Supporting Patient Oriented Research Support Unit and the St Michael's Hospital Foundation). He also received payment for being on a panel at the American Diabetes Association, for a talks at the Toronto Reference Library, for writing a brief in an action for side effects of a drug for Michael F. Smith, Lawyer and a second brief on the role of promotion in generating prescriptions for Goodmans LLP and from the Canadian Institutes of Health Research for presenting at a workshop on conflict-of-interest in clinical practice guidelines. He is currently a member of research groups that are receiving money from the Canadian Institutes of Health Research and the Australian National Health and Medical Research Council. He is member of the Foundation Board of Health Action International and the Board of Canadian Doctors for Medicare. He receives royalties from University of Toronto Press and James Lorimer & Co. Ltd. for books he has written. This does not alter my adherence to PLOS ONE policies on sharing data and materials. There are no restrictions on sharing data.

## Conclusions

Over 18% of the total time between when companies file for drug approval until the drug is available is due to decisions made by companies. Company decision times are shorter for drugs with expedited approvals compared to drugs with standard approvals.

## Introduction

The time between when drug companies file a New Drug Submission (NDS) to get permission to market a new active substance (NAS, a molecule never marketed before in Canada) and when the drug is actually available for purchase has two components. The first is the time taken in the regulatory process until the drug receives a Notice of Compliance (NOC, market authorization). The second is the time between the NOC and when the drug appears on pharmacy shelves. The first period is largely determined by the activities of the Therapeutic Products Directorate (TPD) and the Biologics and Genetic Therapies Directorates (BGTD), the arms of Health Canada that approve small molecule drugs and biologics, respectively [1]. The second period is determined by internal decisions by the company marketing the drug. Some of these decisions may include whether to wait until a recommendation on provincial/territorial funding has been made, how to position the drug with respect to potential competitors, training sales staff in promotion, etc.

Studies of the delay between when companies file a NDS and when drugs reach Canadian patients typically focus on the time in the regulatory review process and do not analyze the company decision time [2, 3]. The lack of differentiation between the two times can potentially lead to a misplaced focus on what types of reforms are necessary should it be necessary to expedite the marketing of new drugs.

This study looks at the length of the two different time periods and the percent of total time from NDS to marketing taken up by company time. Secondarily, it examines whether there is a difference in these time periods for drugs that received a standard review and those that received either of two expedited reviews–priority review or a Notice of Compliance with conditions–review processes designed to ensure that promising therapies for serious, life-threatening or debilitating illnesses reach Canadians in a timely manner [4, 5]. Finally, because companies may wait for a recommendation about public funding before marketing their products, the marketing date is compared to the date when the funding recommendation is made.

## Methods

### Data sources

A list of all NAS approved in Canada between January 1, 2014 and December 31, 2018 was compiled from the annual reports of the TPD and BGTD. (Reports are available by directly contacting the directorates at publications@hc-sc.gc.ca) The brand and generic names of the drugs were recorded in an Excel spreadsheet along with the dates when the companies applied for a NDS, the dates when the drugs received a NOC, whether the drugs received a standard review (300 days) or an expedited review (priority review– 180 days, NOC/c review– 200 days) and whether they were small molecule drugs or biologics. The annual reports are regarded as the authoritative sources of such information. Health Canada does not provide information about when companies apply for new indications to existing products or when it makes decisions about these applications. Therefore, this study only examines NAS and not new indications for drugs already marketed.

Health Canada's Drug Product Database (DPD) [6] gives the date when a product is originally marketed, defined on the website as the "earliest marketed date recorded in the Drug Product Database". The DPD was initially searched on February 7, 2020 and the search was repeated on July 9, 2020 and the marketing date was recorded on the same Excel spreadsheet.

The Common Drug Review (https://www.cadth.ca/about-cadth/what-we-do/products-services/cdr/reports) and the pan-Canadian Oncology Drug Review (https://www.cadth.ca/pcodr/find-a-review), both arms of the Canadian Agency for Drugs and Technology in Health, make publicly available the timelines for the drugs that they review. The date of the final recommendation was recorded.

## Data analysis

Mean times in days over the entire 5-year time period were calculated between NDS and NOC (review time), between NOC and the marketing date (company decision time) and between NDS and the marketing date (total time) for all drugs and then for drugs with a standard review and an expedited review, for small molecule drugs versus biologics and for drugs with a priority review versus those that went through the Notice of Compliance with conditions pathway. The marketing date and the date of the final recommendation on funding were compared to see which one came first. Finally, median times in days for each of review time, company decision time and overall time for the entire sample of drugs were also calculated for individual years. The company decision time as a percent of the total time was calculated for all drugs, drugs with a standard review and drugs with an expedited review. Times were compared between standard and expedited review drugs using a two-tailed t-test. Repeating the analyses using medians and the Mann-Whitney test did not change the outcomes and so only means are reported. Review times, company decision times and overall times over each of the 5 years were separately compared for all drugs using the Kruskal-Wallis test. All calculations were done with Prism 8.3.1 (GraphPad Software).

## Results

From January 1, 2014 to December 31, 2018 there were 176 NAS approved in Canada. The original marketing dates were available for 157 of these and the analysis is based on this group. Over the 5-year period, 83% to 96% of drugs approved were eventually marketed, depending on the year. Ninety-eight drugs had a standard review and 59 had an expedited review. S1 File contains the complete data.

The mean total time for all drugs was 537 days (95% confidence interval (CI) 490, 583), the review time was 403 days (95% CI 374, 433) and the company time was 133 days (95% CI 101, 165) (Table 1). Over 18.5% of the total time was due to company decisions (Table 1). All three times were significantly lower for expedited review drugs versus standard review drugs as was the percent of total time due to company decision– 14.4% (95% CI 11.0, 17.8) versus 21.2% (95% CI 17.6, 24.8), p = 0.0102 (t-test) (Table 1).

The date when the funding recommendation was finalized was available for 114 products. The market date occurred before the funding recommendation date for 104 (91.2%) products and on the same date for one product.

There was no statistically significant difference in median review time (p = 0.2386, Kruskal-Wallis test), median company decision time (p = 0.7226, Kruskal-Wallis test) or median overall time (p = 0.2804, Kruskal-Wallis test) over the 5-year period (Table 2).

Total time and review times were statistically significantly longer for drugs that went through the NOC/c pathway compared to those that received a priority review while company

**Table 1. Time from New Drug Submission to marketing of new active substances.**

| | Time from NDS to marketing (total time) in days (mean, 95% confidence interval) | Time from NDS to NOC in days (review time) (mean, 95% confidence interval) | Time from NOC to marketing (company decision time) in days (mean, 95% confidence interval) | Company decision time as percent of total time (mean, 95% confidence interval) |
|---|---|---|---|---|
| **All drugs** | 537 (490, 583) | 403 (374, 433) | 133 (101, 165) | 18.5 (15.9, 21.1) |
| **Standard review** | 646 (585, 708)* | 467 (429, 505)* | 179 (131, 228)† | 21.2 (17.6, 24.8)‡ |
| **Expedited review** | 354 (313, 395)* | 298 (264, 332)* | 56 (38, 74)† | 14.4 (11.0, 17.8)‡ |

NDS = New Drug Submission

NOC = Notice of Compliance

*Statistically significant difference, p < 0.0001 (t-test)

†Statistically significant difference, p = 0.0002 (t-test)

‡ Statistically significant difference, p = 0.0102 (t-test)

time was not statistically significantly different. All three times were not statistically significantly different when small molecules and biologics were compared. (Data not shown.)

## Discussion

Almost one fifth of the time between when companies file a new drug submission to the time when doctors are able to prescribe a drug is due to company decisions. This figure drops to 13.9% for drugs that received an expedited review compared to 21.5% for drugs with a standard review indicating that companies move more rapidly in making these drugs available. Shorter company decision time could be a reflection of the therapeutic value of the drugs, the revenue that companies expect from sale of the drugs or a combination of both. However, there is no difference in company decision time when biologics are compared to small molecules or when drugs with a priority review are compared to those that went through the NOC/c pathway. Priority review times were shorter than review times in the NOC/c pathway.

In the vast majority of cases, companies marketed their products before the final funding recommendation was made suggesting that the time taken for the funding recommendation was not a factor in delaying marketing. There are no other studies that have explored these two time periods so the Canadian data cannot be compared to data from other countries.

Neither Health Canada's review time nor the company decision time about when to actually put the medicine on pharmacy shelves changed over the 5-years. The absence of any change in these two metrics means that over the time period there was no change in company time as a percent of total time from filing of an NDS to the time the drug was actually marketed.

**Table 2. Review time, company decision time, total time: 2014–2018.**

| Median review time (IQR) (days)* | | | | | Median company time (IQR) (days)† | | | | | Median overall time (IQR) (days)‡ | | | | |
|---|---|---|---|---|---|---|---|---|---|---|---|---|---|---|
| **2014** | **2015** | **2016** | **2017** | **2018** | **2014** | **2015** | **2016** | **2017** | **2018** | **2014** | **2015** | **2016** | **2017** | **2018** |
| 388 (222, 665) | 356 (330, 451) | 349 (280, 406) | 349 (261, 415) | 349 (213, 454) | 41 (15, 102) | 68 (28, 151) | 51 (23, 145) | 62 (31, 113) | 57 (35, 120) | 512 (309, 781) | 479 (368, 820) | 430 (311, 562) | 411 (283, 527) | 405 (307, 588) |

IQR = interquartile range

*No statistically significant difference, p = 0.0947 (Kruskal-Wallis test)

†No statistically significant difference, p = 0.8738 (Kruskal-Wallis test)

‡No statistically significant difference, p = 0.1683 (Kruskal-Wallis test)

The reason behind the time that companies take to make a decision about when to market a drug was not explored in this study but it is probably due to a combination of factors such as distribution of the drug across the country, preparing marketing materials, competition from other similar products and remaining patent time.

Median Canadian review times in 2017 were about 100 days longer than those in the United States, although 70 days shorter than in the European Union [7]. The shorter American time may be a reflection of greater use of expedited review processes or it may be because Health Canada is less efficient than the Food and Drug Administration. Depending on the balance of the two factors there may be an opportunity to shorten Canadian review times and market drugs more quickly. However, this study also shows that drugs could be brought to market faster if companies are able to improve their decision making time once drugs have been approved. Treating standard review drugs in a similar manner to priority review ones would result in them reaching the market 123 days earlier.

This study relied on the accuracy of the information on Health Canada websites and in Health Canada reports and there was no way of verifying the accuracy of that information.

## Conclusion

Over 18% of the total time between when companies file for drug approval until the drug is available is due to decisions made by companies. The time taken for a funding recommendation to be made does not appear to be a factor in when companies decide to market their drugs. Company decision times are shorter for drugs with expedited approvals compared to drugs with standard approvals.

## Supporting information

**S1 File. Complete data.**
(XLSX)

## Author Contributions

**Conceptualization:** Joel Lexchin.

**Data curation:** Joel Lexchin.

**Formal analysis:** Joel Lexchin.

**Methodology:** Joel Lexchin.

**Writing – original draft:** Joel Lexchin.

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
