## [Decision Letter · Decision Letter 0]

27 Aug 2020

PONE-D-20-04228

How long do new medicines take to reach Canadian patients after companies file a submission: a cohort study

PLOS ONE

Dear Dr. Lexchin,

Thank you for submitting your manuscript to PLOS ONE. After careful consideration, we feel that it has merit but does not fully meet PLOS ONE’s publication criteria as it currently stands. Therefore, we invite you to submit a revised version of the manuscript that addresses the points raised during the review process.

We look forward to receiving your revised manuscript.

Kind regards,

Maarten Postma

Academic Editor

PLOS ONE

Journal Requirements:

'In 2016-2019, Joel Lexchin was a paid consultant on two projects: one looking at developing principles for conservative diagnosis (Gordon and Betty Moore Foundation) and a second deciding what drugs should be provided free of charge by general practitioners (Government of Canada, Ontario Supporting Patient Oriented Research Support Unit and the St Michael’s Hospital Foundation). He also received payment for being on a panel at the American Diabetes Association, for a talks at the Toronto Reference Library, for writing a brief in an action for side effects of a drug for Michael F. Smith, Lawyer and a second brief on the role of promotion in generating prescriptions for Goodmans LLP and from the Canadian Institutes of Health Research for presenting at a workshop on conflict-of-interest in clinical practice guidelines. He is currently a member of research groups that are receiving money from the Canadian Institutes of Health Research and the Australian National Health and Medical Research Council. He is member of the Foundation Board of Health Action International and the Board of Canadian Doctors for Medicare. He receives royalties from University of Toronto Press and James Lorimer & Co. Ltd. for books he has written. '

a. Please confirm that this does not alter your adherence to all PLOS ONE policies on sharing data and materials, by including the following statement: "This does not alter our adherence to  PLOS ONE policies on sharing data and materials.” (as detailed online in our guide for authors http://journals.plos.org/plosone/s/competing-interests).  If there are restrictions on sharing of data and/or materials, please state these.

Please note that we cannot proceed with consideration of your article until this information has been declared.

Reviewers' comments:

Reviewer's Responses to Questions

**Comments to the Author**

1. Is the manuscript technically sound, and do the data support the conclusions?

Reviewer #1: Partly

2. Has the statistical analysis been performed appropriately and rigorously? 

Reviewer #1: I Don't Know

3. Have the authors made all data underlying the findings in their manuscript fully available?

Reviewer #1: Yes

4. Is the manuscript presented in an intelligible fashion and written in standard English?

Reviewer #1: Yes

5. Review Comments to the Author

Reviewer #1: Summary of the research and your overall impression

The manuscript claims that the company decision time takes a certain part of the time between the submission of a New Drug Submission (NDS) and first date that it is on the market.

The strength of this manuscript is the use of solely publicly available information. A weakness is that the publicly available information can not be verified. Different dates of ‘Date of NDS submitted’ and ‘Date of Notice of Compliance’ are mentioned in the reports compared to dates mentioned on the website (https://hpr-rps.hres.ca/reg-content/regulatory-decision-summary-result.php?lang=en&term=#).

Discussion of specific areas for improvement

Major issues

1. The author has done extensive research, however something crucial has been missed: The first marketing date in the Health Canada’s Drug Product Database (DPD) is not always the earliest date of marketing. Therefore the ‘data of marketing’ does not always match the earliest date of marketing. I strongly advise the author to check the dates on the earliest date of marketing again, specifically for:

- Praluent;

- Bosulif;

- Invokana;

- Ferriprox;

- Nitisinone (here it appears that the marketing data of MDK-nitisinone was used);

- Ibrance;

- Mictoryl;

- Kevzara, and

- Cosentyx.

2. Although most of the ‘date NDS submitted’ data in the supplementary data are correct, there seems to be a few dates accidentally copied wrong, namely:

- Dymista – 4 November 2013;

- Tremfya – 25 November 2016, and

- Tegsedi – 7 March 2018.

3. Although most of the ‘Date of Notice of Compliance’ data in the supplementary data are correct, however there seems to be one date accidentally copied wrong, namely:

- Ravicti – 18 March 2016.

4. The introduction should be expanded and clarified to ensure that readers understand exactly why this research question is interesting/ why this is a problem.

5. The author is off to a good, interesting start. It is beneficial to this study to add two sub-analyses: 1) the influence of the two expedited reviews: priority review versus Notice of Compliance with conditions; and 2) The difference in time between the TPD and BGTD.

6. While the author appears to have a solid discussion, it appears to me it misses a clear take-home message to the discussion: why is it important to know what the company-decision time is?

Minor issues

1. Although the introduction is clear, no references were used in the first paragraph. The author should add one or more references here (line 56).

2. The author should clarify the definition of internal decisions by the company in the introduction (line 56) to avoid confusion: does this for example include the choice to wait with pricing regulation/ submitting to the HTA body till after receipt of NOC or NOC/c?

3. The author should move the aim of the paper to the last paragraph of the introduction. Subsequently should the aim also be addressed in the discussion.

4. A good addition to the methods section would be to specify that this study only looks at New Active Substances and not at extensions of the indication.

5. The author’s mentioning of the time which a standard review and an expedited review should take (line 76 and 77) creates the idea that the time from NDS to NOC is too long. This should be clarified/discussed in the discussion. The extra time can (partly) be explained by the time the regulatory authority has to wait on the additional data from the company to make up for the deficiency in the application.

6. The author should add a reference to the TPD and BGTD reports (line 73).

7. The author should add abbreviations under Table 1.

8. The discussion should be expanded and be put into context of previously published research on this topic: Is the time in regulatory process better or worse in comparison to these articles?

9. The author should add a reference to line 119 and 124.

- Unfortunately, I do not have the expertise to consider the statistics.

Although this paper needs some revisions, I really appreciate the issue the author poses: the time between the submission of the NDS till the drug is available for the patients depends not only on the regulatory authority, but also on the decision(s) of the company. If this issue is better defined this will be a great publication which is easy to read and to reproduce!

6. PLOS authors have the option to publish the peer review history of their article (what does this mean?). If published, this will include your full peer review and any attached files.

Reviewer #1: No

---

## [Author Response · Author response to Decision Letter 0]

29 Aug 2020

August 29, 2020

To the Editor:

Thank you and the reviewer for the comments and the opportunity to revise my manuscript. Below I indicate how I have responded to the various comments.

Reviewer #1: Summary of the research and your overall impression

The manuscript claims that the company decision time takes a certain part of the time between the submission of a New Drug Submission (NDS) and first date that it is on the market.

The strength of this manuscript is the use of solely publicly available information. A weakness is that the publicly available information can not be verified. Different dates of ‘Date of NDS submitted’ and ‘Date of Notice of Compliance’ are mentioned in the reports compared to dates mentioned on the website (https://hpr-rps.hres.ca/reg-content/regulatory-decision-summary-result.php?lang=en&term=# [hpr-rps.hres.ca]).

The fact that the information cannot be verified is already mentioned in the manuscript.

It is now stated that the annual reports from the TPD and the BGTD are regarded of the authoritative sources of information about the date of NDS and NOC for individual products.

Discussion of specific areas for improvement

Major issues

1. The author has done extensive research, however something crucial has been missed: The first marketing date in the Health Canada’s Drug Product Database (DPD) is not always the earliest date of marketing. Therefore the ‘data of marketing’ does not always match the earliest date of marketing. I strongly advise the author to check the dates on the earliest date of marketing again, specifically for:

- Praluent;

- Bosulif;

- Invokana;

- Ferriprox;

- Nitisinone (here it appears that the marketing data of MDK-nitisinone was used);

- Ibrance;

- Mictoryl;

- Kevzara, and

- Cosentyx.

I am not sure what other database is available to check the date of marketing but if the reviewer can supply that information, I would be happy to look at the marketing dates in that database. In the meantime, I have checked the DPD for the drugs that the reviewer names. All of the dates listed in the DPD were correctly recorded except for those for Nitisinone and Cosentyx and those have been corrected.

2. Although most of the ‘date NDS submitted’ data in the supplementary data are correct, there seems to be a few dates accidentally copied wrong, namely:

- Dymista – 4 November 2013;

- Tremfya – 25 November 2016, and

- Tegsedi – 7 March 2018.

The dates of NDS submitted for these three drugs were checked against the dates given in the annual reports from the TPD and BGTD and the dates have been corrected.

3. Although most of the ‘Date of Notice of Compliance’ data in the supplementary data are correct, however there seems to be one date accidentally copied wrong, namely:

- Ravicti – 18 March 2016.

The date has been corrected.

4. The introduction should be expanded and clarified to ensure that readers understand exactly why this research question is interesting/ why this is a problem.

The following sentence has been added to the Introduction: “The lack of differentiation between the two times can potentially lead to a misplaced focus on what types of reforms are necessary should it be necessary to expedite the marketing of new drugs.”

5. The author is off to a good, interesting start. It is beneficial to this study to add two sub-analyses: 1) the influence of the two expedited reviews: priority review versus Notice of Compliance with conditions; and 2) The difference in time between the TPD and BGTD.

These additional comparisons have been done and the results are reported.

6. While the author appears to have a solid discussion, it appears to me it misses a clear take-home message to the discussion: why is it important to know what the company-decision time is?

The following passage has been inserted into the Discussion based on the reviewer’s comment: “Median Canadian review times in 2017 were about 100 days longer than those in the United States, although 70 days shorter than in the European Union.(6) The shorter American time may be a reflection of greater use of expedited review processes or it may be because Health Canada is less efficient than the Food and Drug Administration. Depending on the balance of the two factors there may be an opportunity to shorten Canadian review times and market drugs more quickly. However, this study also shows that drugs could be brought to market faster if companies are able to improve their decision making time once drugs have been approved. Treating standard review drugs in a similar manner to priority review ones would result in them reaching the market 123 days earlier.”

Minor issues

1. Although the introduction is clear, no references were used in the first paragraph. The author should add one or more references here (line 56).

A reference has been inserted.

2. The author should clarify the definition of internal decisions by the company in the introduction (line 56) to avoid confusion: does this for example include the choice to wait with pricing regulation/ submitting to the HTA body till after receipt of NOC or NOC/c?

The following section has been added to the Introduction: “Some of these decisions may include whether to wait until a recommendation on provincial/territorial funding has been made, how to position the drug with respect to potential competitors, training sales staff in promotion, etc.”

3. The author should move the aim of the paper to the last paragraph of the introduction. Subsequently should the aim also be addressed in the discussion.

The aim has been moved to the end of the Introduction. The Discussion already begins with a statement of the finding about how long companies take, on average, to make their marketing decisions.

4. A good addition to the methods section would be to specify that this study only looks at New Active Substances and not at extensions of the indication.

The following passage has been added to the end of the first paragraph in the Methods section: “Health Canada does not provide information about when companies apply for new indications to existing products or when it makes decisions about these applications. Therefore, this study only examines NAS and not new indications for drugs already marketed.”

5. The author’s mentioning of the time which a standard review and an expedited review should take (line 76 and 77) creates the idea that the time from NDS to NOC is too long. This should be clarified/discussed in the discussion. The extra time can (partly) be explained by the time the regulatory authority has to wait on the additional data from the company to make up for the deficiency in the application.

I respectfully disagree with the reviewer on this point. The length of time for the various review processes is just a statement of fact and there is no implication that these times are too long. The Summary Basis of Decision document lists the dates between when a Screening Deficiency Notice is issued, if there is one, and when a Screening Acceptance Letter is issued, but does not appear to give the times used in the review process waiting for responses to Clarification Requests. Therefore, the total amount of time taken up waiting for companies to respond to Health Canada requests is not known. As a result, I have not made any changes with respect to this point.

6. The author should add a reference to the TPD and BGTD reports (line 73).

These reports are not posted on the Health Canada website but the email address to request them is already provided in the manuscript.

7. The author should add abbreviations under Table 1.

NDS and NOC are spelled out below Table 1.

8. The discussion should be expanded and be put into context of previously published research on this topic: Is the time in regulatory process better or worse in comparison to these articles?

The Discussion already notes that there is no previously published research comparing review times and company decision times in other countries. The new reference 7 briefly mentions review times in the United States and the European Union.

9. The author should add a reference to line 119 and 124.

“(Table 1)” was added to the end of the sentence that started on line 119 (now line 157) to indicate the source of the figures cited in the preceding sentence. “(Table 1)” already appeared at the end of the sentence on line 124 (now line 159) to indicate the source of the figures cited in the preceding sentence. 

- Unfortunately, I do not have the expertise to consider the statistics.

Although this paper needs some revisions, I really appreciate the issue the author poses: the time between the submission of the NDS till the drug is available for the patients depends not only on the regulatory authority, but also on the decision(s) of the company. If this issue is better defined this will be a great publication which is easy to read and to reproduce!

I thank the reviewer for the compliment.

---

## [Editor Report · Decision Letter 1]

7 Oct 2020

How long do new medicines take to reach Canadian patients after companies file a submission: a cohort study

PONE-D-20-04228R1

Dear Dr. Lexchin,

We’re pleased to inform you that your manuscript has been judged scientifically suitable for publication and will be formally accepted for publication once it meets all outstanding technical requirements.

Kind regards,

Maarten Postma

Academic Editor

PLOS ONE
---

## [Editor Report · Acceptance letter]

22 Oct 2020

PONE-D-20-04228R1 

How long do new medicines take to reach Canadian patients after companies file a submission: a cohort study 

Dear Dr. Lexchin:

I'm pleased to inform you that your manuscript has been deemed suitable for publication in PLOS ONE. Congratulations! Your manuscript is now with our production department. 

Kind regards, 

on behalf of

Dr. Maarten Postma 

Academic Editor

PLOS ONE